# A Survey on Personalized and Pluralistic Preference Alignment in Large Language Models

**Zhouhang Xie**[1], **Junda Wu**[1], **Yiran Shen**[1], **Raghav Jain**[1], **Yu Xia**[1], **Xintong Li**[1],
**Aaron Chang**[2], **Ryan Rossi**[3], **Tong Yu**[3], **Sachin Kumar**[4], **Bodhisattwa Prasad Majumder**[5]
**Jingbo Shang**[1], **Prithviraj Ammanabrolu**[1], **Julian McAuley**[1]

[1]University of California, San Diego    [2]University of California, Los Angeles
[3]Adobe Research    [4]The Ohio State University    [5]Allen Institute for AI
{zhx022,jw069,jes038,r6jain,yux078,xil240,prithvi,jmcauley}@ucsd.edu
aaronchang21@g.ucla.edu, {ryrossi, tyu}@adobe.com
kumar.1145@osu.edu, bodhisattwam@allenai.org

## Abstract

Personalized preference alignment for large language models (LLMs), the process of tailoring LLMs to individual users' preferences, is an emerging research direction spanning the area of NLP and personalization. In this survey, we present an analysis of works on personalized alignment and modeling for LLMs. We introduce a taxonomy of preference alignment techniques, including training time, inference time, and additionally, user-modeling based methods. We provide analysis and discussion on the strengths and limitations of each group of techniques and then cover evaluation, benchmarks, as well as open problems in the field.

## 1 Introduction

Recently, significant progress has been made in aligning LLMs to the *overall* preferences of users (Zhao et al., 2024; Shen et al., 2023). However, prior works show that there is no one-size-fits-all solution for preference alignment (Sorensen et al., 2024; Kirk et al., 2024a). Intuitively, while there are universal preferences that are shared across users, such as "it is good to respond in the tone of a friendly and helpful assistant", user preferences are often also individualized and use-case dependent (i.e., contextual). For example, users might have different preferences towards the tone and style of responses (Jang et al., 2023), and even for the same user, the preferred style of response would change depending on the context of interaction, akin to prior works in contextualized personalization (Meng et al., 2023). For example, an expert might have different preferences from a beginner when asking the same question about a concept, and even the same person might have different preferences depending on when they interact with the system.

Along with the emergence of these challenges, the notion of personalization in the context of NLP and LLM research has recently attracted increasing attention within the NLP and machine learning communities (Sorensen et al., 2024; Kirk et al., 2024a; Flek, 2020). At first glance, personalization seems to be an intuitive solution for catering LLMs to individualized and contextual preferences described above. However, in practice, the notion of personalization for LLMs is convoluted, ranging from applications of LLMs to classical personalization tasks (Tan & Jiang, 2023; Chen et al., 2024c) to role-playing and simulation of individual human behaviors (Chen et al., 2024d; Mou et al., 2024). As we shall discuss later (Section 7), not all types of personalization benefit individual users in a conversation setting.

In this work, we focus on personalized and pluralistic preference alignment, a specific notion of personalization that aims at adapting an LLM's behavior to dynamic user *preferences* across individuals, groups, and contexts to enhance user satisfaction. We start by defining the problem formulation for personalized preference alignment (Section 3), then introduce an intuitive technique taxonomy covering training and test-time methods for preference

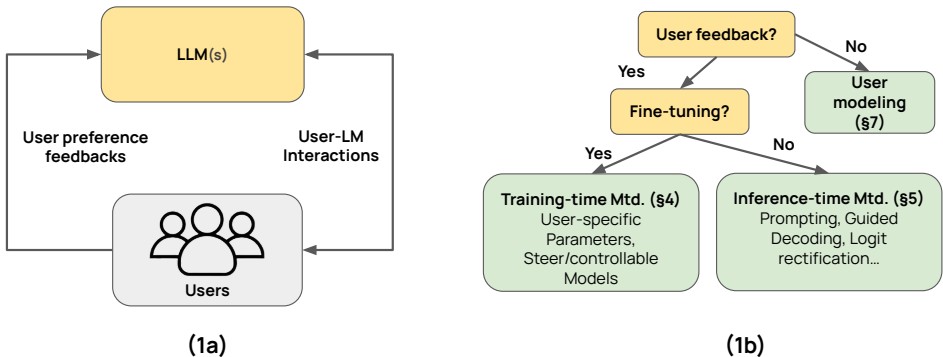

Figure 1: (1a) Overview on personalized preference alignment for LLMs. This includes training (Section 4) and test-time (Section 5) methods, leveraging various feedbacks such as verbal feedback and choices. (1b) An over-simplified decision tree for determining the class of method to use for personalized preference alignment.

alignment (Section 4, Section 5). We then discuss the connection between user modeling and personalized preference alignment (Section 7), and discuss benchmark and evaluation (Section 6) as well as open problems in the field (Section 8).

**What's covered?** Broadly speaking, as shown in Figure 1, personalization in the context of preference alignment can be categorized into two classes of methods: Training-time (Section 4) and inference-time (Section 5) personalization that leverages personalized user feedback at different stages of LLMs' life cycle. Additionally, when user preference feedback is absent, personalized adaption of LLMs and LLM-based systems can sometimes also be achieved by modeling the user, which we discuss in Section 7.

**What's not covered.** Under the theme of LLMs and personalization, there are other relevant areas such as user-behavior modeling (i.e., systems that predict the behavior of users) (Tan & Jiang, 2023; Wu et al., 2024b; Chen et al., 2024c), user-group-behavior modeling (i.e., LLM role-playing) (Chen et al., 2024d;d; Tseng et al., 2024), and LLMs *for* personalized recommender systems (e.g., LLMs as a component in product recommender systems) (Tan & Jiang, 2023; Wu et al., 2024b; Chen et al., 2024c). However, these lines of work does not involve directly catering the behavior of LLMs or LLM-based conversational systems to user preferences. We point interested readers to related surveys (Section 2) on these topics.

**Contribution statement.** Despite numerous recent efforts on consolidating works that involve LLM and personalization (Sorensen et al., 2024; Kirk et al., 2024a; Zhang et al., 2024b; Wu et al., 2024a; Tan & Jiang, 2023; Wu et al., 2024b; Chen et al., 2024c), we show personalized alignment is an emerging valuable research direction, with its own methods and evaluation paradigm. To this end, this survey provides a comprehensive overview of personalized and pluralist alignment while establishing its differences and connections to adjacent domains. Further, personalized preference alignment is an emerging domain with no universally acknowledged evaluation and benchmarks. To make it easier for future research to access existing evaluation schemes, we provide an overview of the current state of evaluation methods. Overall, this survey serves as an up-to-date resource for practitioners and researchers working on personalized and pluralistic alignment and, more broadly, LLM personalization and alignment.

## 2 Related Surveys and Key Differences

**User modeling and personalizing LLMs to simulate user behavior.** Recently, a few position papers have discussed the concept of personalized preference alignment (Sorensen et al., 2024; Kirk et al., 2024a). However, these works focus on discussing the implications without summarizing current progress. There have also been surveys on personalizing LLMs (Zhang et al., 2024b; Wu et al., 2024a), but they do not differentiate between the replicating user

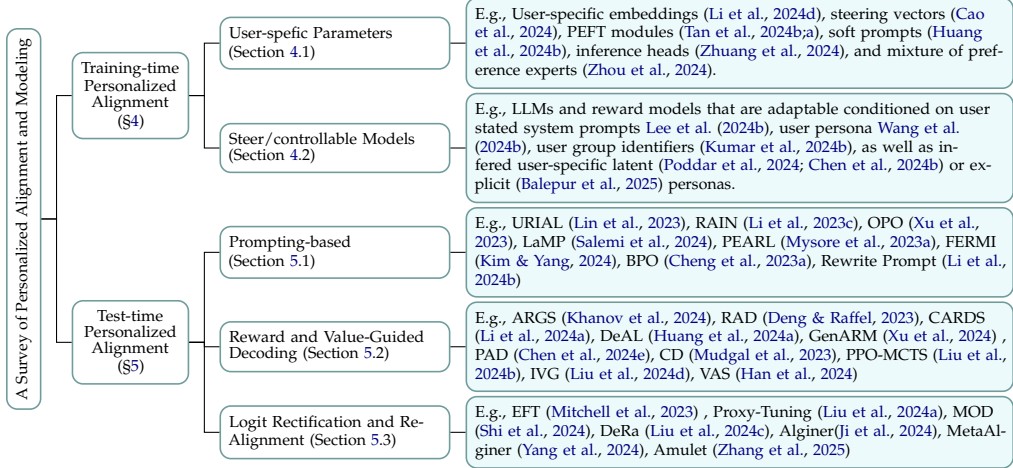

Figure 2: Technique taxonomy on personalized and pluralistic preference alignment.

behavior to catering for user preferences. Finally, another stream of recent works focuses on LLM-based role-laying (Tseng et al., 2024). However, their focus is still on empowering LLMs to replicate certain user groups' behavior. In contrast, our work focuses on summarizing the progress with respect to personalized preference alignment.

**Application of LLMs to downstream personalization tasks.** There are also a few surveys on the *application* of LLMs to other tasks related to personalization, such as recommender systems and user modeling (Tan & Jiang, 2023; Wu et al., 2024b; Chen et al., 2024c), personalized wearable devices (Li et al., 2024e), personalization on the web (Chen et al., 2024c), narrative-drive recommendation (Mysore et al., 2023b), and LLM-based role-laying (Chen et al., 2024d), which are different from the scope covered in this work since the goal of these lines of research is not to build an optimal LLM-based dialogue system. We provide further clarification of our scope in Section 3, and discuss in more detail the relationship between (personalized) user modeling and preference alignment in Section 7.

## 3 Problem Formulation and Techniques Taxonomy

### 3.1 Problem Statement

We begin by introducing the notion of personalization in the context of preference alignment, as illustrated in Figure 1-a. In this setting, system deployers aim to build an LLM (or LLM-based system) that caters specifically to each individual user's unique preferences.

For clarity, let $\mathcal{X}$ denote the input space (e.g., user queries or contextual prompts), and $\mathcal{Y}$ denote the output space (e.g., responses generated by the LLM). We formalize the personalized reward function as

$$r : \mathcal{X} \times \mathcal{Y} \times \mathcal{U} \to \mathbb{R},$$

which measures how well an LLM's response $y$ to an input $x$ satisfies the unique preferences of user $u$. Our goal is to learn a set of individualized policies $\{\pi_u\}_{u \in \mathcal{U}}$, where each policy $\pi_u$ is tailored exclusively to user $u$. Such an objective can be expressed as:

$$\pi_u^* = \arg\max_{\pi_u} \mathbb{E}_{x \sim \mathcal{X}, y \sim \pi_u(\cdot|x,u)} \left[ r(x, y, u) \right] \quad \forall u \in \mathcal{U} \tag{1}$$

In other words, for each user $u$, the optimal policy $\pi_u^*$ is one that, in expectation over the distribution of inputs, generates responses that best satisfy that user's individual preferences. However, compared to personalization which emphasizes individualized prediction for each user, approaches for achieving pluralistic alignment are more diverse, sometimes involving systems that produce sets of outputs catering for different preferences. Meanwhile, the LLM-based policies can often have shared parameters, or are built from a single model conditioned on different user information, as we shall discuss in Section 4 and Section 5.

### 3.2 Technique Overview

The problem formulation discussed above yields a straight-forward way to partition existing methods, as shown in Figure 2. Specifically, in order to obtain a LLM-based policy that cater to each individual's preferences, it is crucial to be able to adapt the LLM itself depending on the interacting user. Naturally, this can be achieved both by *training* models with user-specific parameters (Section 4.1) or making the model steerable with respect to user inputs (Section 4.2), as shown in Figure 1-b. On the other hand, fine-tuning or adapting LLMs is frequently costly. This challenge motivates another category of works that aims at influencing a pre-trained LLMs' behavior at inference time, with methods such as prompting (Section 5.1), controlled-decoding (Section 5.2), and logit manipulation (Section 5.3). Additionally, we note that there are personalization techniques that nevertheless improve base LLMs towards the goal of catering to individual preferences without using user feedback, such as building user-specific memories (e.g., (Yuan et al., 2025; Zhang et al., 2022)) following the assumption that users generally likes to be remembered. We provide discussion of this complementary class of valuable techniques in Section 7.

### 3.3 Granularity of Personalization

While personalized preference alignment aims to cater to, ultimately, individualized preferences, personalization typically suffers from the issue of sparse feedback (Li et al., 2023b). Specifically, an individual user's interactions with the system are frequently too few to allow meaningful learning to happen. To this end, similar to works in adjacent personalization areas such as recommender systems (Li et al., 2023b), it is often helpful to exploit the fact that there can be *groups* of users that shares similar preferences, thus bypassing the feedback sparsity issue. Similar to these prior works, personalized LLM alignment can also happen on different granularity: individual users' levels and user groups based on user profiles and social relationships (Sorensen et al., 2024; Kumar et al., 2024b). We note that there is also a special case of "contextual" shared preference between user groups, where a set of users momentarily shares preferences based on their purpose of interacting with LLMs. For example, a group of users that are seeking help from an LLM-based therapy chatbot may collectively wish the LLM to act as a helpful therapist Stade et al. (2024). We provide discussions on this special case in Section 7.

## 4 Training-time Personalized Alignment

To effectively adapt LLMs and LLM-based systems to personalized user preferences, a straight-forward solution is to develop specialized models via training. In this section, we introduce two popular classes of techniques: building models with user-specific parameters and training models that are sensitive to input user preferences.

### 4.1 Learning from Feedbacks with User-specific Parameters

**Motivation.** One of the most straightforward ways to build LLM that caters to individualized user preference is by keeping separate parameters for each user (or user group), effectively learning a set of policies whose behavior slightly deviates from each other without explicitly specified user preference under the standard RLHF setting.

**Comparative Analysis.** Addressing the challenge of inferring user-specific preferences without requiring explicit specification, (Li et al., 2024d) employs a lightweight user model to learn from human feedback, while (Cao et al., 2024) follows a similar setting, but learn per-user steering-vectors to achieve personalization. Following similar intuition, (Tan et al., 2024b) and (Tan et al., 2024a) advocates for a dedicated per-user PEFT module that stores individual behavior patterns. Other than adapter modules, (Huang et al., 2024b) explores a complimentary parameter-efficient personalization approach via soft prompts. (Zhuang et al., 2024) introduces HYDRA, a factorization framework that couples a shared base model with user-specific heads, yielding notable improvements over prompt-based methods. (Zhou et al., 2024) introduces RLPHF, which merges outputs from specialized

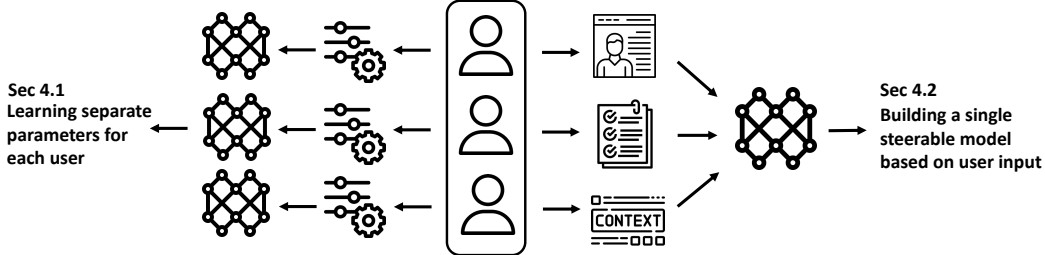

Figure 3: Personalized language model alignment during training time.

expert LLMs using a lightweight Preference Control Model (PCM) that dynamically adjusts token predictions based on user context; i.e., mixture of preference experts. Finally, in contrast to learning user-specific parameters in LLMs, (Park et al., 2024a) explored learning multiple reward models to handle the trade-off between bias and variance in personalized preference alignment.

**Limitations.** Despite these advances, existing personalized RLHF frameworks often struggle to simultaneously ensure personalized adaptation and global model performance, with many approaches relying on complex multi-stage processes or additional components that may hinder model scalability (Park et al., 2024b; Han et al., 2024; Lee et al., 2024a). Furthermore, challenges remain in robustly handling heterogeneous and strategic feedback while integrating efficient privacy-preserving techniques, pointing to the need for more streamlined and resilient solutions such as federated personalized alignment (Zhang et al., 2024a; Jiang et al., 2024).

## 4.2 Building Steerable Model that Adapts Responses Based on User Input

**Motivation.** Another popular choice for personalization is building a single base model that's steerable (Sorensen et al., 2024), where are *single* models' behavior changes based on the user it is interacting with, similar to prior research for controlled text generation (Hu et al., 2017). This line of research often assumes user inputs relevant to individualized preferences are available, such as explicitly stated preferences or personas that imply user preferences are available at inference time, bypassing the need for user-specific parameters.

**Comparative Analysis.** Following the assumptions that user can provide their own preferences as prompts to LLM, (Lee et al., 2024b) train an instruction-following LLM that can adapt to user-written values in system prompt via data synthesis, (Wang et al., 2024b) builds base models geared towards llm-as-a-judge (Zheng et al., 2023) use-cases that can adapt to explicitly stated user personas. Aside from building steerable LLM policies, such adaptability can also be built into reward models. For example, (Pitis et al., 2024) builds context-conditioned reward models, which are then used for personalized (i.e., context-conditioned) alignment. Similarly, (Kumar et al., 2024b) builds reward models conditioned on user group identifiers on Reddit to achieve personalization. We note that there are works that attempt to infer latent user representations when users don't explicitly state their preference, but still build steerable models based on these latent representations. For example, (Balepur et al., 2025) infer user persona from choices using LLMs, and train model to adapt to those personas. In contrast, (Chen et al., 2024b) adopts a plurality-based approach, using ideal point and mixture modeling to learn a common latent preference space that generalizes to new users, Finally, (Poddar et al., 2024) also learns user latent factors, but instead opts to use user-conditional reward models to achieve personalization.

**Limitations.** While alleviating the need for maintaining user-specific parameters, this class of methods often depends on user such as personas and verbal preferences, which are unavailable in standard RLHF settings. Further, collecting datasets with diverse user personas itself is poses a challenge, which we discuss extensively in Section 6.

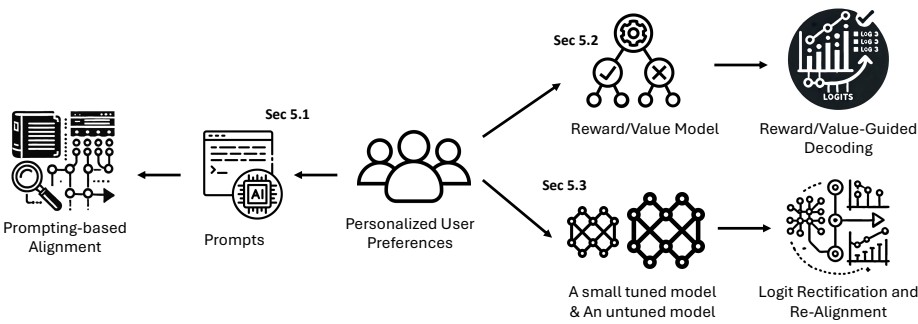

Figure 4: Personalized language model alignment during test time.

# 5 Test-time Personalized Alignment

The growing need for language models that can adapt on the fly to diverse user preferences—while keeping compute costs manageable and simplifying deployment—has spurred a variety of test-time alignment techniques. These methods adjust the decoding process, without altering the underlying model parameters, to steer outputs toward desired behaviors and make debugging easier. We broadly categorize these approaches into three groups: prompting and context optimization, reward- and value-guided decoding methods, and rectification-based decoding and correction.

## 5.1 Prompting-based Alignment Methods

**Motivation.** Prompting-based approaches modify the input context—via in-context examples, retrieval, or prompt rewriting—to elicit behavior that mirrors specific user preferences. This training-free strategy is particularly attractive for personalization, as it can quickly adapt to new or evolving user profiles (see Figure 4).

**Comparative Analysis.** A central challenge for prompting-based methods is achieving personalized alignment without the need for full-scale fine-tuning. (Lin et al., 2023) and (Li et al., 2023c) propose methods that harness a small number of stylistic examples or self-correction mechanisms to nudge LLMs toward outputs that align with individual user styles. Similarly, (Xu et al., 2023) dynamically incorporate external memories to retrieve rules tailored to diverse social norms, while (Salemi et al., 2024) and (Mysore et al., 2023a) further personalize responses by retrieving user-specific items to augment the prompt. In parallel, prompt optimization techniques by (Kim & Yang, 2024), (Cheng et al., 2023a), and (Li et al., 2024b) iteratively refine prompts based on misaligned outputs, thereby progressively incorporating user feedback.

**Limitations.** Despite their promise for personalization, these methods are limited by the fixed context length and increased computational overhead when handling complex, dynamic user profiles. Additionally, the reliance on relatively static representations of user preferences can hinder real-time adaptation to rapidly changing individual needs.

## 5.2 Reward and Value-Guided Decoding Methods

**Motivation.** Reward and value-guided decoding methods integrate personalized alignment objectives directly into the LLM decoding by adjusting token probabilities based on reward signals or value functions. This approach supports fine-grained, token-level personalization and enables LLMs to adapt their outputs on the fly to meet individual user preferences.

**Comparative Analysis.** One of the major challenges is balancing high-reward personalized output with maintaining natural language fluency. (Khanov et al., 2024) and (Deng & Raffel, 2023) propose frameworks that incorporate reward signals into the decoding process, which can be tuned to reflect personalized quality metrics. Building on this, (Li et al., 2024a) and (Huang et al., 2024a) introduce strategies that break the generation process

into segments, allowing for iterative refinement toward personalized and fluent outputs. Complementing these reward-guided methods, (Xu et al., 2024) and (Chen et al., 2024e) focus on scalable, test-time alignment by employing autoregressive reward models and personalized reward modeling, respectively. In parallel, value-guided techniques offer another path to personalization: (Mudgal et al., 2023) and (Liu et al., 2024b) integrate auxiliary value functions during decoding, whereas (Liu et al., 2024d) and (Han et al., 2024) demonstrate that combining implicit and explicit value guidance can further enhance the model's responsiveness with respect to users' personalized preferences and values.

**Limitations.** A key limitation of reward and value-guided methods is their reliance on pre-trained reward or value models, which introduces challenges such as limited transparency and sensitivity to adversarial inputs. Additionally, the need for real-time evaluation of reward signals and value functions often results in significant computational overhead, making it challenging to scale these methods for high-throughput, low-latency applications.

### 5.3 Logit Rectification and Re-Alignment Approaches

**Motivation.** Logit rectification approaches modify the internal decision process of LLMs by integrating corrective signals—often from smaller, fine-tuned models or auxiliary modules—directly into the decoding stage. This strategy enables personalized re-alignment without retraining the entire model, thereby offering a lightweight path to tailor outputs to individual user needs.

**Comparative Analysis.** A key challenge in this category is to harness the complementary strengths of large pretrained models and smaller, aligned models. (Mitchell et al., 2023) introduce emulated fine-tuning, which decouples the contributions of pre-training and fine-tuning by combining the knowledge of a large model with the behavioral adjustments of a small model; a special case, LM up-scaling, demonstrates that ensembling can emulate the effects of full fine-tuning without additional training. In a similar vein, (Liu et al., 2024a) propose proxy-tuning, a lightweight algorithm that steers the output distribution of a large black-box model using the difference between the predictions of a tuned small model and its untuned counterpart. Addressing the need for multi-objective personalization, (Shi et al., 2024) develop a method that computes a linear combination of predictions from several base models, enabling flexible adjustment of competing objectives, while (Liu et al., 2024c) propose decoding-time re-alignment to dynamically control the degree of alignment without retraining. Complementing these strategies, (Ji et al., 2024) offers a model-agnostic correction approach that learns residuals to refine outputs, and (Yang et al., 2024) extends this idea with a generalizable framework that supports multi-objective and personalized alignment by adjusting target objectives via prompt updates. Finally, (Zhang et al., 2025) frame re-alignment as an online per-token optimization problem with a closed-form solution, achieving real-time adaptation to diverse and evolving user preferences.

**Limitations.** Rectification-based approaches are highly dependent on the quality and consistency of the auxiliary correction signals. Their effectiveness can be limited if the small, tuned models do not capture the full complexity of user-specific nuances, which may lead to a mismatch between the corrective adjustments and the large model's inherent output tendencies. Additionally, ensuring robust performance across a wide range of personalization scenarios without introducing extra latency remains a practical challenge.

## 6 Dataset & Evaluation

**Datasets.** Early attempts on constructing datasets for personalized preference alignment typically rely on data synthesization, with some recent attempts in collecting real-world data for preference alignment. We provide an overview of relevant benchmarks and evaluations in Table 1. For example, (Cheng et al., 2023b) introduce the Domain-Specific Preference (DSP) dataset by augmenting the Alpaca instruction corpus (Taori et al., 2023) with answers targeting different domain (e.g., Academia or Entertainment). (Jang et al., 2023) introduce Personalized Soups (P-SOUP), where Alpaca instruction prompts (Taori et al., 2023) are answered by an LLM in contrasting styles along three dimensions (expertise level, ver-

| Reference | Task/Data | Metric (Dimension) | Notes |
|---|---|---|---|
| DSP (Cheng et al., 2023b) | Prompt selection | PPA (Domain: Acad., Bus., Ent., Lit. & Art) | Tailored prompts |
| P-SOUP (Jang et al., 2023) | Pairwise feedback | PWR (Expertise, Info., Friendliness) | GPT-4 simulated |
| MULTIFACETED (Lee et al., 2024b) | Preference alignment | AAS (Human pref.) | Diverse preferences |
| HH-RLHF (Bai et al., 2022) | RLHF dataset | CRS (Helpfulness, Harmlessness) | Personalized dimensions |
| HelpSteer2 (Wang et al., 2024d) | RLHF feedback | MFS (Helpfulness, Harmlessness, Humor) | Humor-enhanced |
| PRISM (Kirk et al., 2024b) | Demographic alignment | AFI (Fairness across demographics) | Participatory alignment |
| LaMP (Salemi et al., 2024) | Personalized generation | POQ (User-profile retrieval) | Retrieval-augmented |
| LongLaMP (Kumar et al., 2024a) | Personalized long-text generation | POQ (User-profile retrieval) | Retrieval-augmented |
| PersonalLLM (Zollo et al., 2024) | Personalization benchmark | PAS (Individual prefs.) | Beyond uniform alignment |
| ALOE (Wu et al., 2024c) | Multi-turn dialogues | PCS (Persona consistency) | Persona-specific dataset |
| PersoBench (Afzoon et al., 2024) | Persona-aware dialogue | PAA (Persona awareness) | Zero-shot evaluation |

Table 1: Compressed overview of datasets and benchmarks for personalized alignment and generation.

bosity, and tone) and ranked based on predefined style preferences. On a much larger scale, (Lee et al., 2024b) constructed the Multifaceted Collection dataset, containing about 65k instructions, each paired with three LLM-generated responses that vary along thousands of dimensions. Finally, (Zollo et al., 2024) present PersonalLLM, an open benchmark that generate preference rankings from a set of simulated synthetic "users". In parallel, other recent works have assembled personalized human preference datasets to study personalization with real users . The PRISM Alignment dataset (Kirk et al., 2024b) is a notable example, recording 8,011 live chat interactions between 1,500 users across 75 countries, with each user's persona profile and preference feedbacks (ratings), providing a complementary setting for evaluating personalized preference alignment.

**Evaluation in Personalized Alignment.**   Evaluating personalization in LLM alignment requires measuring how well a model's output matches the particular preferences of a user. When the evaluation dataset includes response variations across known dimensions (e.g., style, tone), evaluation often uses multi-dimensional scoring or pairwise comparisons. For example, (Jang et al., 2023) assess their proposed methods by scoring responses along each dimension separately, but assumes different users place different value on the set of dimensions. Another popular evaluation approach is to employ LLM-as-a-judge Zheng et al. (2023), but prompt the judge LLMs with pre-defined user personas (Wu et al., 2024c; Lee et al., 2024b). This evaluation formulation leads to the creation of specialized evaluation models such as PerSE (Wang et al., 2024b), a 13B Llama-2 based evaluator fine-tuned to judge alignment with personal profiles. Overall, a unique open challenge in personalized alignment evaluation is realistically simulating personalized preference judgment (Zheng et al., 2023; Wu et al., 2024c; Lee et al., 2024b; Wang et al., 2024b), calling for further investigations.

**Limitations.**   Despite progress on model evaluations catered for personalized preference alignment, most assessments rely heavily on rule-based metrics or persona-based LLM-as-a-judge Zheng et al. (2023), yielding simulations of user satisfaction. Since these heuristics and personas for evaluations are use-case specific, there are currently no unified evaluation across studies, posing a challenge for systematic evaluation and progress tracking. As the research community moves toward widely accepted multidimensional preference benchmarks and the development of publicly available evaluators, more consistent and comparable metrics are expected to emerge.

# 7   On Personalized User Modeling and Personalized Preference Alignment

**Why this section?** As discussed previously in Section 2, personalized preference alignment is a subset of personalization research for LLMs. Specifically, there is another emerging research area that focuses on LLM-based user modeling Tan & Jiang (2023), building LLM-based *simulations* for individual users, which most commonly manifest as directly predicting users' responses. While prior surveys do not differentiate between personalized user modeling and preference alignment (Zhang et al., 2024b; Wu et al., 2024a; Tseng et al., 2024), we note that these are two distinct and complementary research directions. To this end, this section serves two purposes. First, we clearly distinguish between the two

research directions currently studied under LLMs and personalization, making it easier for future researchers to sift through relevant literature. Second, we discuss various ways user modeling can enable better-personalized preference alignment.

**Personalized preference alignment research benefits from personalized user modeling.** Due to the difficulty in collecting large-scale feedbacks from real users, current research for personalized preference alignment relies heavily on user simulation. For example, various recently proposed datasets rely on persona-grounded simulation with LLMs to build benchmarks (Cheng et al., 2023b; Jang et al., 2023; Lee et al., 2024b; Zollo et al., 2024) or evaluations (Zheng et al., 2023; Wu et al., 2024c; Lee et al., 2024b; Wang et al., 2024b) for personalized preference methods. To this end, better simulation of diverse, realistic user behaviors naturally improves the development of better personalized alignment methods.

**Modeling individual user (sometimes) enables preference-free personalized alignment.** Meanwhile, by modeling individual users, system deployers can still increase user satisfaction, even without directly modeling user preferences. For example, in areas such as personalized review prediction (Xie et al., 2023; Ni et al., 2019) and recommender system (Wu et al., 2024b), user behavior happens to strongly correlate with user preferences, and thus better prediction of user behavior directly improves preference alignment. Similarly, related research directions such as LLM-based chit-chat dialogue systems with personalized user memory (e.g., (Yuan et al., 2025)) also better caters to the preference of individual users by remembering personal facts, even when there are no explicit modeling of user preferences. Finally, simulating specific desired personas such as teachers (Wang et al., 2024c), therapists (Stade et al., 2024), and travel-planners (Chen et al., 2024a) also enables LLMs to better cater to the corresponding user groups, such as students, patients, and travelers.

# 8 Future Works and Emerging Directions

**Online and Continuous Personalized Alignment** While existing work on personalized preference alignment primarily explores the setting of learning user preference from offline data or given explicitly stated user preference, another complementary setting is personalized LLM alignment in an online setting (Chen et al., 2024f). Additionally, given prior success in adjacent research (such as user modeling) on continuous personalization over multiple dialogue sessions (Li et al., 2024c; Zhang et al., 2023; Zhong et al., 2024; Qian et al., 2024), personalized preference alignment in a multi-session dialogue setting is a natural extension, which typically models turn-wise or user-provided preference feedback. Nevertheless, recent work Zhao et al. (2025) show large language models frequently fail to recall user preference in continuous personalization setting, calling for more elaborate solution for personalization of LLMs over time.

**Addressing long and complex user-generated value statements** As discussed in prior sections, personalized preference alignment in LLMs frequently relies on instruction following ability of LLMs as building blocks for alignment methods, both at training time and inference time. However, recent works show long and complex instruction-following is still an open challenge (Wu et al., 2024d; Gavin et al., 2024). Given the prevalence of personalized alignment methods that rely on explicit verbal preference statements (Section 4.2 and Section 5.1), it is still unclear whether existing methods can support complex and long user value statements. To this end, developing benchmarks and methods to further research the instruction-following ability of LLMs on complex user preference value statements can help LLM-based dialogue systems better handle rich, multifaceted user preferences.

# 9 Conclusions

In this survey, we perform a comprehensive analysis of existing methods, datasets, and benchmarks for personalized preference alignment in LLMs and LLM-based dialogue systems. We discuss various classes of methods and their advantages and drawbacks, covering both training and inference-time, as well as user-knowledge-based personalized preference alignment methods. We also discuss limitations and future directions for LLMs to cater to diverse and individualistic preferences.

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

## A  On Risk of Pluralistic Preference Alignment

We note that as discussed in prior works (Kirk et al., 2024a), naively personalizing LLMs to individualized preference can raise other issues such as amplifying bias or inducing polarization. To this end, similar to adjacent domain in personalization such as recommender system Wang et al. (2023), ensuring safety and fairness in pluralistic alignment remain an continuously important and open challenge. Meanwhile, pluralistic alignment generally requires collecting diverse preference signals from users, calling for privacy-aware method such as differentially private LLM alignment Goel et al. (2025).

## B  On Datasets and Benchmarks for Pluralistic Alignment

Unlike recent work on general preference alignment for LLMs—which spans training and inference time methods, *inter alia* (Rafailov et al., 2023; Wang et al., 2025; Wu et al., 2025; Kong et al., 2024; Song et al., 2025), a unique ongoing challenge in the field is the lack of commonly acknowledged benchmarks and evaluation methods. To this end, we show a few recent works and their evaluation datasets in Table 2, and hope this makes accessing relevant dataset easier and facilitate comparisons on shared benchmarks.

| Method | Dataset |
|---|---|
| RAG/PAG (Salemi et al., 2024) | LAMP (Salemi et al., 2024) |
| OPPU (Tan et al., 2024b) | LAMP (Salemi et al., 2024) |
| Personalized-PCs (Tan et al., 2024a) | LAMP (Salemi et al., 2024) |
| HyDRA (Zhuang et al., 2024) | LAMP (Salemi et al., 2024) |
| P-DPO (Li et al., 2024d) | Prism Alignment Kirk et al. (2024b) |
| BiPO (Cao et al., 2024) | AI Persona (Perez et al., 2023), TruthfulQA (Lin et al., 2022), ADVBench (Zou et al., 2023) |
| SPT (Huang et al., 2024b) | ConvAI2 (Dinan et al., 2019) |
| System Message Gen. (Lee et al., 2024b) | Multi-faceted Benchmark and other helpfulness/harmlessness benchmarks (Lee et al., 2024b) |
| PerSE (Wang et al., 2024a) | Per-MPST (Wang et al., 2024a), Per-DOC (Zhu et al., 2023) |
| CARM (Pitis et al., 2024) | RPR Dataset (Pitis et al., 2024) |
| PIPT (Balepur et al., 2025) | Beavertails/SHP (Ji et al., 2023), Stanford Preferences (Ethayarajh et al., 2022), HH-RLHF (Bai et al., 2022), Mnemonic (Balepur et al., 2024) |
| PAL (Chen et al., 2024b) | Anthropic Persona Perez et al. (2023), Pick-a-Pic (Kirstain et al., 2023) |
| PEARL (Mysore et al., 2023a) | WORKSM, AITA, see PEARL paper |
| FERMI (Kim & Yang, 2024) | OpinionQA (Santurkar et al., 2023), GlobalOpinionQA Durmus et al. (2024), LAMP Salemi et al. (2024) |
| Prompt Rewrite (Li et al., 2024b) | FtPersLlm (Li et al., 2023a) |
| Linear Align. (Gao et al., 2024) | HH-RLHF Bai et al. (2022) |

Table 2: Inexhaustive List of Recent Personalized/Pluralistic Alignment Methods and Datasets Used

