# OpenReview forum: "A Survey on Personalized and Pluralistic Preference Alignment in Large Language Models"
_colmweb.org/COLM/2025/Conference — COLM 2025_

### Official Review · Reviewer_Voas · 2025-05-11

**Rating:** 6
**Confidence:** 3
**Ethics Flag:** 1

**Summary:**

This paper is a survey that presents a comprehensive overview of personalized and pluralistic preference alignment in Large Language Models (LLMs). Personalized preference alignment has emerged as a key research direction, motivated by the observation that users often have diverse stylistic preferences when interacting with personal AI systems. The authors begin by categorizing existing methods into training-time and test-time approaches. Training-time methods include strategies that either fine-tune LLMs with user-specific parameters or construct steerable models capable of adapting their responses based on user input. In contrast, test-time methods modify the decoding process without altering the underlying model weights. These are further classified into three main categories: prompting and context optimization, reward- and value-guided decoding, and logit rectification and correction. Additionally, the paper explores user modeling as a complementary strategy for personalization. The survey also reviews available datasets and benchmarks, emphasizing the limited availability of real user data and the widespread reliance on synthetic or simulated user personas. It notes that evaluation practices in this area remain fragmented and lack standardized protocols.

**Reasons To Accept:**

The authors present a thorough and valuable survey on the timely and increasingly important topic of personalized and pluralistic preference alignment in Large Language Models (LLMs).

**Reasons To Reject:**

This paper is a survey and does not include any original experimental contributions. As such, its technical novelty is limited. To offer more substantial insights to the community, it would have been beneficial to include empirical comparisons across different methods in varied settings. While the survey is well-organized and informative, its suitability for a technical conference may be debatable given the lack of new experimental findings.

==================

Considering that COLM accepted survey papers last year, this paper would also be a suitable candidate for acceptance.

---

> ### Author Response · Authors · 2025-06-01
>
> Thank you for your time in reviewing our survey. Regarding the points you raised, we'd like to clarify the following --
>
> - [W1] Re. experiments.
>   - Thanks for mentioning this. However, we’d like to clarify that our work is a survey paper, and thus, our focus in this work is not empirical experiments. To this end, similar survey papers in the prior iterations of COLM share our scope, e.g. [1-4] in prior surveys in COLM (see below).
>   - Meanwhile, personalized/pluralistic alignment is an emerging research direction with no consensus on best evaluation dataset practices (Table 1), making direct comparisons between methods challenging. Regardless, we will add Table 1 and relevant discussions to the paper to provide further information about the relationship between current works to our readers.
>
> Prior surveys in COLM:
>
> [1] LLM as a Mastermind: A Survey of Strategic Reasoning with Large Language Models, COLM 24
>
> [2] What Are Tools Anyway? A Survey from the Language Model Perspective, COLM 24
>
> [3] Beyond Accuracy: Evaluating the Reasoning Behavior of Large Language Models - A Survey, COLM 24
>
> [4] A Survey on Deep Learning for Theorem Proving, COLM 24

---

> > ### Comment · Reviewer_Voas · 2025-06-06
> >
> > Thank you for your response. I would like to confirm that I have carefully read your reply.

---

### Official Review · Reviewer_oXrd · 2025-05-11

**Rating:** 7
**Confidence:** 4
**Ethics Flag:** 1

**Summary:**

The authors provided a systematic survey on preference alignment with personality settings, including many works that depend on training or customized LLM inference workflows.

**Reasons To Accept:**

It is an excellent survey covering various aspects of personalized preference alignment, including training & inference techniques, and evaluation, including datasets or benchmarks.

I also enjoy the authors' clear formalization of this topic, as well as distinguishing it from related areas such as LLM role playing and listing corresponding surveys, which makes the paper highly reader-friendly.

**Reasons To Reject:**

1. Despite the above strengths, it is sad to see that the authors did not clarify how this topic was different from other problems in definition or experimental setting, as certain works involved in this survey can be shared in the aspect of method design.

2. The authors focused on analyzing specific works and techniques. However, it would further strengthen the survey if they could provide deeper insight into this problem in Sections 1 & 3.

3. The authors are recommended to add references to more related works to make this survey better, including:
    1. Fine-tuning methods like the famous DPO[1] and recent works[2, 3].
    2. Test-time methods like RE-CONTROL[4] and ICDPO[5].

[1] Rafailov, Rafael, et al. "Direct preference optimization: Your language model is secretly a reward model." Advances in Neural Information Processing Systems 36 (2023): 53728-53741.

[2] Wang, Zekun Moore, et al. "PopAlign: Diversifying Contrasting Patterns for a More Comprehensive Alignment." arXiv preprint arXiv:2410.13785 (2024).

[3] Wu, Junkang, et al. "RePO: ReLU-based Preference Optimization." arXiv preprint arXiv:2503.07426 (2025).

[4] Kong, Lingkai, et al. "Aligning large language models with representation editing: A control perspective." Advances in Neural Information Processing Systems 37 (2024): 37356-37384.

[5] Song, Feifan, et al. "Instantly Learning Preference Alignment via In-context DPO." Proceedings of the 2025 Conference of the Nations of the Americas Chapter of the Association for Computational Linguistics: Human Language Technologies (Volume 1: Long Papers). 2025.

---

> ### Author Response · Authors · 2025-06-01
>
> Thank you for your valuable insight to our work. Regarding the points raised --
>
> - [W1] Despite the above strengths, it is sad to see that the authors did not clarify how this topic was different from other problems in definition or experimental setting, as certain works involved in this survey can be shared in the aspect of method design.
>   - Thanks, compared to other personalization tasks (e.g., replicating user behavior), while some evaluations transfer to this line of work, a common distinct setting/evaluation is preference-ranking with LLM as a judge mimicking user or user groups with specific preferences, typically conditioned on synthetic persona or preferences [e.g., 5, 6, 8, 11, 12, 14, 17, 18].
>   - However, as discussed in section 6 and shown in Table 1, currently, evaluation in the field varies significantly, and we hope our work can accelerate convergence of our community on evaluating personalized preference alignment, by delineating relevant works.
>   - We will add this discussion to section 6, and point to the discussion in problem setting.
>
> [W2] The authors focused on analyzing specific works and techniques. However, it would further strengthen the survey if they could provide deeper insight into this problem in Sections 1 & 3.
>   - Please see the response to W1, where we will add a discussion about the uniqueness of the problem setting compared to adjacent fields and provide further discussion re. the problem itself.
>
> [W3] The authors are recommended to add references to more related works.
>   - Thanks. We will make reference to the suggested general methods and other further related work as fit.

---

> > ### Comment · Reviewer_oXrd · 2025-06-04
> > **Response to the authors**
> >
> > Response from the authors has tackled my issues, so I raise my score. Good luck!

---

### Official Review · Reviewer_9asZ · 2025-05-13

**Rating:** 6
**Confidence:** 4
**Ethics Flag:** 1

**Summary:**

This paper presents a comprehensive survey on personalized and pluralistic preference alignment in large language models (LLMs). It categorizes existing techniques into training-time, inference-time, and user-modeling-based methods, and offers valuable insights into datasets, evaluation practices, and open challenges. I find this to be a well-structured, timely, and valuable survey. While some sections could benefit from further clarification or synthesis, the paper overall meets the standard for acceptance.

**Reasons To Accept:**

1. Timely and relevant topic
The alignment of LLMs to user-specific preferences is an emerging but rapidly growing area. This paper addresses a clear gap in existing literature by focusing specifically on preference alignment, not just general personalization or user modeling.

2. Comprehensive taxonomy
The classification into training-time, inference-time, and user-modeling techniques is intuitive and clearly explained. The paper nicely distinguishes between parameter-based personalization (e.g., PEFT modules, soft prompts) and controllable generation approaches.

3. Clear problem formulation
The formalization of the personalization problem using a user-conditioned reward function is rigorous and helps position the surveyed methods within a consistent framework.

**Reasons To Reject:**

1. Limited synthesis across methods
While many methods are described in detail, the paper sometimes reads like a catalog rather than a deeply analytical synthesis. For example: What are the trade-offs between training-time and inference-time personalization in real deployment scenarios?

2. Ambiguity in scope definitions.
The term pluralistic preference alignment is introduced early, but its distinction from personalized alignment is somewhat vague. Does “pluralistic” simply refer to accommodating diversity across users, or does it imply multi-objective or group-aware alignment?

3. Minor writing issues. Some repeated citations (e.g., “Chen et al., 2024c” appears multiple times for different topics). Typos like “user-spefic” instead of “user-specific” in Section 4.1.

---

> ### Author Response · Authors · 2025-06-01
>
> Thank you for providing valuable feedback to our work. Regarding your comments/concerns:
>
> - [W1] Limited synthesis across methods While many methods are described in detail, the paper sometimes reads like a catalog rather than a deeply analytical synthesis. For example: What are the trade-offs between training-time and inference-time personalization in real deployment scenarios?
>   - Thanks - while we share your view that the community would benefit from head-to-head comparison between various methods, we wanted to point out that due to the emerging nature of the field, current works are frequently parallel to each other and employ different benchmarks, baselines, and evaluations [Table 1, 1-18].
> Regardless, current works show (a) personalized methods are better than non-personalized ones (e.g., DPO, SFT, Zero-shot) [5, 8, 11, 18]; and (b) training-time interventions are generally more effective [e.g., 2, 3, 4, 12, 14]. We will add this discussion along with Table 1 to the appendix.
> [W2] Ambiguity in scope definitions. The term pluralistic preference alignment is introduced early, but its distinction from personalized alignment is somewhat vague. Does “pluralistic” simply refer to accommodating diversity across users, or does it imply multi-objective or group-aware alignment?
>   - Thanks. We want to clarify that pluralistic and personalized preference alignment are working towards the same general goal, i.e., building a system that exhibits behavior that is preferable to each individual user (section 3.1).
> While the two terms differs slightly in pitch, e.g., pluralistic alignment emphasizes “building a system that satisfies diverse preferences across the whole user population”, the term “personalization,” i.e., “for each user, being able to cater to his/her preference” actually pitches towards the same objective from a different angle.
>   - We’ll add this comment to section 3.1 to highlight this.
> - [W3] Minor writing issues..
>   - Thanks for the catch, we will clear the citations and fix the typos accordingly.

---

### Official Review · Reviewer_ckzC · 2025-05-19

**Rating:** 5
**Confidence:** 4
**Ethics Flag:** 1

**Summary:**

This survey explores the topic of personalized preference alignment in LLMs, organizing techniques into training-time methods (user-specific parameters or steerable models) and inference-time approaches (prompting, reward-guided decoding, and logit rectification). The paper distinguishes preference alignment from user modeling, reviews evaluation, benchmarks, and identifies future challenges in continuous personalization and handling complex user value statements.

**Questions To Authors:**

1. Reading through the sections 4 and 5 it is not clear to the reader which methods are better performance-wise or efficiency-wise and where are the gaps. Could you expand on that?
2. In the future work directions, what are the anticipated challenges and potential ways to address them?
3. What are the ethical and privacy concerns that you foresee with personalized preference alignment?

**Reasons To Accept:**

- Provides a clear taxonomy of methods for personalized preference alignment that makes them more accessible and provides structure into studying this topic.
- Explains the motivations behind existing methods, compares them, and lists their key limitations.
- Unlike prior studies, it differentiates personalized preference alignment from related personalization areas which helps clarify their boundaries.

**Reasons To Reject:**

- The comparative analysis lacks focus on performance and efficiency aspects of existing methods on different benchmarks.
- There is no dedicated discussion of ethical, privacy or other potential risks and implications of personalized preference alignment.
- Discusses limitations and a few future work directions but does not elaborate on them. It would be useful to further discuss the quality gaps and challenges that remain to be addressed in this field.

---

> ### Author Response · Authors · 2025-06-01
>
> Thank you for your valuable feedback. Please find the responses to each individual comments as follows:
>
> - [W1] The comparative analysis lacks focus on performance and efficiency aspects of existing methods on different benchmarks.
>   - Thanks for mentioning this point. While we share your view that head-to-head comparison of methods on a standard set of benchmarks is important, we wanted to clarify that, as we discussed in the paper (line 62+, introduction; line 304+, limitations of dataset & evaluation), since personalized/pluralistic alignment is an emerging field, there are current no universally acknowledged benchmarks yet, making a direct comparison between methods challenging.
>   - To make this more concrete, we select representative methods mentioned in our work and summarize their evaluation benchmarks and baselines in Table 1. As shown in the table, while [1-4] conduct evaluation on the LAMP benchmark, benchmark selection generally exhibits great diversity [e.g., 5-18]
>   - For methods that are evaluated on the same benchmark (e.g., LAMP), contemporary works often compare with the same set of baselines (e.g., RAG/PAG proposed as baseline in the LAMP benchmark paper itself) due to being developed in parallel. Based on self-reported performances, Per-PCs [3] is a more efficient method than OPPU [2]; while performance-wise it outperforms Hydra [4] on some subtasks while being lower on others. We will add this information, relevant discussions here, and Table 1 to the appendix of the paper.
> As we discussed in section 6, it would be beneficial for our community to converge towards unified benchmarks for comparison between methods (as you mentioned). To this end, part of the motivation/contribution of this work is to clearly delineate the status of personalized preference alignment (which itself is an emerging field), to help our community to build upon each others’ work and accelerate the progress towards unified evaluation/comparisons.
> - [W2] There is no dedicated discussion of ethical, privacy or other potential risks and implications of personalized preference alignment.
>   - Thanks. We will highlight this more explicitly in the final version, and add a discussion for relevant matters such as bias and privacy in the appendix, so readers can easily trace the relevant literature (currently in related work section, line 70-71, kirk2024a). Please see response to Q3 for further details.
> - [W3] Discusses limitations and a few future work directions but does not elaborate on them. It would be useful to further discuss the quality gaps and challenges that remain to be addressed in this field.
>   - Thanks. For Continuous personalized alignment (future work paragraph 1), recent work [Zhao ICLR 25] shows LLMs frequently fail to correctly remember and apply user preferences, even when the preference is explicitly stated.
>   - To this end, better long and complex instruction-following models (future work paragraph 2) seems a natural remedy, though, as mentioned, there are limited data and methods geared towards addressing this problem in personalized alignment settings, and it would be interesting to see if current success such as data synthesis [e.g., 8 in Table 1] holds.
>   - We will add these discussions to the future directions section.
> - [Q1] Reading through sections 4 and 5 it is not clear to the reader which methods are better performance-wise or efficiency-wise and where are the gaps. Could you expand on that?
>   - Please see response to W1, and general response to reviewers.
> - [Q2] In the future work directions, what are the anticipated challenges and potential ways to address them?
>   - Please see response to W2.
> - [Q3] What are the ethical and privacy concerns that you foresee with personalized preference alignment?
>   - Personalized/pluralistic alignment in LLMs naturally requires collecting user-specific information such as user profiles and personal descriptions and/or their behavioral data. To this end, this direction shares privacy concerns with other adjacent domains, such as recommender systems. Similarly, personalization can limit individual users’ access to diverse content, similar to the filter bubble effect in other personalization domains.
>   - As mentioned in response to W3, we will add a dedicated discussion to these issues.

---

### Author Response · Authors · 2025-06-01
**General response to reviewers**

Thank you for your time and valuable feedback. We are encouraged to hear that reviewers find our survey well-written/thorough (9asz, oxrd, voas), our taxonomy/problem formulation clear/comprehensive (9asz, ckcz, oxrd), our topic timely and relevant (9asz) and clearly differentiates from prior surveys (ckcz, oxrd). Reviewers also appreciated our discussion on motivation and limitations of methods (ckzc) and our summary on evaluation/datasets (9asZ, oXrd).

Regarding direct comparisons between methods, we note that as shown in Table 1 (and discussed throughout the paper ((line 62+, introduction; line 304+, limitations of dataset & evaluation), personalized/pluralistic alignment is an emerging field, and thus existing works have significant varying evaluation settings and baselines, which typically involve non-personalized counterparts (e.g., DPO, SFT, ICL). This makes head-to-head comparison challenging from a survey's perspective. To this end, we hope this survey, via outlining relevant works, can help our community converge towards a unified (and better) evaluation and make access to relevant works easier.

Regardless, from reported results from the literature, personalized methods generally outperform non-personalized methods, and training-time methods generally outperform test-time methods. This can also be confirmed from a few works [1-4] that evaluate on LAMP. We will add relevant discussion and Table 1 to the appendix of our work.

Please find more details and responses to other comments in individual comments in the corresponding sections.

---

> ### Author Response · Authors · 2025-06-01
>
> Table 1: Representative methods and their setting
>
> | Method | Paper | Year | Dataset | Automated Evaluation Used | Beats |
> |--------|-------|------|---------|------------|-------|
> | RAG/PAG | 1. [Lamp: When LLMs Meet Personalization](https://arxiv.org/abs/2304.11406) | 2023 | LAMP | Accuracy, RMSE, F1, BLEU | Non-personalized baselines |
> | OPPU | 2. [Democratizing LLMs via Personalized Parameter-Efficient Fine-tuning](https://arxiv.org/abs/2402.04401) | 2024 | LAMP | Accuracy, RMSE, F1, BLEU | RAG and variants |
> | Personalized-PCs | 3. [Personalized Pieces: Efficient Personalized LLMs](https://arxiv.org/abs/2406.10471) | 2024 | LAMP | Accuracy, RMSE, F1, BLEU | RAG and variants, OPPU (more efficient) |
> | HyDRA | 4. [HYDRA: Model Factorization for Black-Box LLM Personalization](https://arxiv.org/abs/2406.02888) | 2024 | LAMP | Accuracy, RMSE, F1, BLEU | ICL, RAG |
> | P-DPO | 5. [Personalized language modeling from personalized human feedback](https://arxiv.org/abs/2402.05133) | 2024 | Prism Alignment | LLM-based evaluation | DPO |
> | BiPO | 6. [Personalized Steering of LLMs via Bi-directional Preference Optimization](https://arxiv.org/abs/2406.00045) | 2024 | Anthropic Eval, TruthfulQA, JainBreaking | LLM-based evaluation | CAA, Free-form Activation Steering |
> | SPT | 7. [Selective Prompting Tuning for Personalized Conversations](https://arxiv.org/abs/2406.18187) | 2024 | ConvAI2 | BLEU, ROUGE, F1, BertScore, Dist | Prompt Tuning |
> | System Message Gen. | 8. [Aligning to thousands of preferences via system message generalization](https://arxiv.org/abs/2405.17977) | 2024 | Multi-faceted Benchmark | LLM-based | Various LLMs |
> | PerSE | 9. [Learning Personalized Alignment for Evaluating Open-ended Generation](https://arxiv.org/abs/2310.03304) | 2023 | Per-MPST, Per-DOC | BLEU, ROUGE, BART | Matrix factorization, LLMs |
> | CARM | 10. [Improving Context-Aware Preference Modeling for LMs](https://arxiv.org/abs/2407.14916) | 2024 | RPR Dataset + Benchmarks | Agreement, Accuracy | Prometheus-2, UltraRM, RM variants |
> | PIPT | 12. Improving Personalization in Preference Tuning...* | 2025 | Beavertails, Stanford Preferences, HHH, Mnemonic | LLM-based evaluation | Few-shot, SFT, DPO |
> | PAL | 13. [PAL: Pluralistic Alignment Framework](https://arxiv.org/abs/2406.08469) | 2024 | Reddit TL;DR, Pick-a-Pic | Task-specific | Non-personalized baselines |
> | PEARL | 14. [Pearl: Personalizing LLM Writing Assistants](https://arxiv.org/abs/2311.09180) | 2023 | WORKSM, AITA | ROUGE, BertScore, LLM-based evaluation | Prompting, RAG |
> | FERMI | 15. [Few-shot Personalization of LLMs with Mis-aligned Responses](https://arxiv.org/abs/2406.18678) | 2024 | OpinionQA, GlobalOpinionQA, LAMP | Accuracy | Prompting variants |
> | Prompt Rewrite | 16. [Learning to Rewrite Prompts for Personalized Text Generation](https://dl.acm.org/doi/10.1145/3589334.3645451) | 2023 | FtPersLlm | BLEU, ROUGE | FtPersLlm baseline |
> | Linear Align. | 17. [Linear Alignment](https://arxiv.org/abs/2401.11458) | 2024 | Personal Preference Eval | LLM-based evaluation | PPO, DPO, Best-of-N |
> | Amulet | 18. [Amulet: ReAlignment During Test Time](https://openreview.net/forum?id=f9w89OY2cp) | 2024 | PPE, HelpSteer, UltraFeedback | LLM-based evaluation | Prompting, Beam Search, Linear Align. |
>
> * Hyperlinking to [12]'s Arxiv site seems to break openreview table rendering.

---

### Decision · Program_Chairs · 2025-07-08

**Decision:**

Accept

**Comment:**

There is a lingering question here whether a survey is a proper place for a conference such as COLM. Assuming it is appropriate, I believe this paper is a good fit for the conference and mostly ready for publication. The authors properly addressed the questions/comments by the reviewers and seem to have covered a large body of literature in their survey.